# Effects of Commercial Polysaccharides Stabilizers with Different Charges on Textural, Rheological, and Microstructural Characteristics of Set Yoghurts

**DOI:** 10.3390/foods11121764

**Published:** 2022-06-15

**Authors:** Zhiwen Ge, Dongjie Yin, Zhiyu Li, Xiaohong Chen, Mingsheng Dong

**Affiliations:** College of Food Science and Technology, Nanjing Agricultural University, Nanjing 210095, China; zhiwenge603@163.com (Z.G.); 2019108007@njau.edu.cn (D.Y.); lizhiyu1008@163.com (Z.L.); xhchen@njau.edu.cn (X.C.)

**Keywords:** polysaccharides, set yoghurt, physicochemical properties, rheology, microstructure

## Abstract

The study investigated the preparation of set yoghurts by adding three common commercial polysaccharide stabilizers, namely sodium alginate (SA), gellan gum (GG), and konjac gum (KGM), in milk fermentation to evaluate their effects on the texture, rheology, and microstructure of set yoghurts. The physicochemical properties, water-holding capacity (WHC), texture, low-field nuclear magnetic resonance (LF-NMR), rheology, and microstructure of set yoghurts added with different kinds and quantities of polysaccharides were compared and analyzed. The results showed that the set yoghurts added with anionic polysaccharide GG had more obvious effects on improving WHC, firmness, and rheological properties compared with the set yoghurt added with KGM and SA. The firmness of set yoghurts with 0.02% (*w*/*v*) GG increased from 1.17 N to 1.32 N, which significantly improved the gel structure. The transverse relaxation time (T_2_) of set yoghurts added with GG was the closest to that of the control. Compared with the set yoghurts added with 0.02% SA and KGM, the free water area (A_23_) of the one added with 0.02% GG decreased most significantly. Moreover, all samples showed shear-thinning behavior, and the apparent elastic and viscous modulus (G′, G″) increased with the increase of GG concentration. The G′ and G″ of set yoghurts with 0.005% SA and KGM were higher than those in the control, decreased when adding 0.010%, and then increased with the increase of SA and KGM. Additionally, the microscopic observation demonstrated that the addition of GG in set yoghurts significantly promoted the formation of larger protein clusters and showed a tighter and more uniform protein network comparing with the other two polysaccharides (SA, KGM).

## 1. Introduction

Yoghurt is a widely-consumed fermented dairy food internationally renowned for its nutritional value of high protein and calcium contents and healthy benefits to human bodies [1,2]. Generally, the strength of pure protein gels formed by casein micelles was commonly low and susceptible to syneresis. Many reports have pointed out that the addition of polysaccharides had a fine effect on improving the structure of yoghurt [3,4]. Presently, several commercialized polysaccharides have been added into the yoghurt production to promote the fermentation process, strengthen the casein network, and reduce dehydration, which mainly includes an anionic polysaccharide, such as gellan gum (GG) [5,6], sodium alginate (SA) [7], xanthan gum [2], pectin [8], etc., and neutral polysaccharide, such as konjac glucomannan (KGM) [1], etc.

Many studies have investigated the effect of polysaccharides as stabilizers on the texture of yoghurt. Zhang et al. [9] reported that anionic polysaccharides produced by *Streptococcus thermophilus* ST1 exhibited the ability to improve textural and microstructural properties of fermented skim milk. Pang et al. [10] reported that anionic polysaccharides—such as GG, SA, which belonged to microbial polysaccharide and seaweed polysaccharide, respectively—could present electrostatic interaction with positive charges on the surface of casein micelles after being added into yoghurt preparation. Moreover, other anionic polysaccharides, such as xanthan gum, alginate, and pectin, have also been reported to improve the textural properties of set yoghurt depending on the associative interactions between polysaccharides and casein micelles [11,12]. Additionally, neutral polysaccharides KGM added in low-fat and skimmed yoghurts cause significant effects on the decrease of syneresis and spontaneous whey separation. However, the comparison of the effects of using commercial polysaccharide stabilizers with different charges for yoghurt in preparing and evaluating their influence on the physicochemical, textural, rheological, and microbiological properties of the set yoghurts remains unclear.

The traditional quality measurements to investigate the textural properties of yoghurts mainly include whey separation, sedimentation, viscosity, etc. [13,14,15], whereas such methodologies often cause yoghurts matrix destruction prior to or during the determination. As a non-destructive and non-invasive technique, the low-field nuclear magnetic resonance (LF-NMR) has been widely utilized for the investigation of the entrapment of water in dairy products, which seemed to be a promising method [11,16]. The relaxometry (T_2_) of water was determined via the transverse relaxation of ^1^H protons [17]. Furthermore, owing to the protective effect of cryo-scanning electron microscopy (cryo-SEM) on the fine microstructure of set yoghurts, it has been widely utilized to study the microstructure of yoghurts with additional stabilizers [18,19]. With the advantages of both visualizing and chemically differentiating dairy products components through specific protein, fats, or polysaccharides stains, confocal laser scanning microscopy (CLSM) has also been widely used to observe microstructure of dairy products. The tryptophan residues in milk could be considered intrinsic fluorescent probes for proteins [20,21].

The aim of the study was to evaluate the influence of three commonly used polysaccharide stabilizers such as SA, GG, and KGM on the texture, rheology, and microstructure of set yoghurts. Therefore, the gelling characteristics of prepared set yoghurts were demonstrated using texture analyzer, LF-NMR, and rheometer. Moreover, the microstructure of different set yoghurts was analyzed by cryo-SEM and CLSM. The interaction mechanism between polysaccharides with different charges and caseins in the set yoghurt system was preliminarily explored.

## 2. Materials and Methods

### 2.1. Materials

Pasteurized cow milk (3.2% *w*/*v* protein, 3.8% *w*/*v* fat and 4.8% *w*/*v* carbohydrate per 100 mL) culture was purchased from Yili Industrial Group Co., Ltd. (Inner Mongolia, China). The commercial SA (98% purity, consisted of L-guluronate and D-mannuronate residues at a molar ratio of 1:1.6, 1.63 × 10^5^ Da), GG (99% purity, consisted of a linear random copolymer of 1,3-glucose residue, 1,4-glucose residue, 1,3-glucuronic acid residue and 1,4 rhamnose residue, 5.00 × 10^5^ Da), and KGM (99% purity, consisted of a linear random copolymer of 1,4-linked-D-glucopyranose and D-mannopyranose units in a molar ratio of 1:1.6, 3.56 ×10^5^ Da) with analytical pure level used in this study were purchased from McLean Biochemical Technology Co., Ltd. (Shanghai, China).

### 2.2. Set Yoghurts Preparation

The anionic polysaccharide SA and GG, and the neutral polysaccharide KGM was selected as experimental treatments to observe the effect of charge carrying capacity of polysaccharides on the texture and structure of fermented set yoghurts. According to the viscosity of polysaccharides, the amount of polysaccharides added in coagulated fermented milk was set [22]. The viscosities of SA, GG, and KGM (1 mg/mL) were determined to be 15.11 mPa·s, 16.73 mPa·s, and 14.46 mPa·s, respectively. According to the multiple relation, their addition ranges were set at 0.005–0.02%, 0.005–0.02%, and 0.005–0.02% (*w*/*v*), respectively.

The ultraviolet sterilization was performed for the pasteurized milk medium for 30 min before inoculation. The inoculation amount of commercial starter (*Lactobacillus bulgaricus* and *Streptococcus thermophilus*) was set at 0.2% (*w*/*v*) concentration, which was then fermented at 42 °C for 6 h in incubator to prepare set yoghurts and was then put in 4 °C overnight for after-ripening to set as the control [21]. The polysaccharides (SA, GG, KGM) were added into milk medium separately and slowly based on the steps of the control group prepared, then mechanically stirred until dissolved completely, and fermented at 42 °C for 6 h in incubator, which were set as the experimental groups. 

### 2.3. Physical Properties, Water-Holding Capacity (WHC), and Texture of Set Yoghurts

The titratable acidity (TA) and WHC values of set yoghurts were determined by direct titration method and centrifugal method, respectively [21,23]. The TA value was expressed in °T. The pH value was measured by a pH Meter (PB-10, Sartorius, Germany), the total solids (TS) were determined by constant temperature drying method with 105 °C, and the total soluble solids (TSS) was determined by refractometer (Recht LXT-500, Shenzhen, China). Subsequently, the textural properties determination of prepared set yoghurts was measured by a Texture Analyzer (Washington, DC, USA). The firmness and gel rupture distance were measured [24,25]. 

### 2.4. LF-NMR Determination

LF-NMR measurement was carried out by a NMI 20 low-field nuclear magnetic resonance imaging analyzer (Niumag Corporation, Shanghai, China) [13]. A 1.5 mL glass vial containing 2.00 g sample was embedded into a 15 mm NMR tube, then inserted into the NMR probe, and the transverse relaxation time (TZ) signal was collected by Carr–Purcell–Meiboom–Gill (CPMG) sequence. The parameter of resonance frequency was 40 MHz, sampling frequency was 200 kHz, sampling frequency was 90°, hard pulse RF pulse width 11 µs, and 180° hard pulse RF pulse width 17 µs. The number of repeated sample was 16, the waiting time of repeated sampling was 3000 ms, and the number of echoes was 18,000 [21].

### 2.5. Characterization of Rheological Properties

The rheological properties determination was according to the method of Ren et al. [26], and steady shear and oscillatory frequency sweep determination was carried out by Discovery HR-10 rheometer system (TA Instrument, New Castle, DE, USA). The gap was set at 500 μm. The steady state shear scanning ranged from 0.1 s^−1^ to 100 s^−1^. Besides, the elastic modulus (G′) and viscous modulus (G″) were measured with fixed 0.1% strain, and frequency scanning ranged from 0.1–10 Hz in the dynamic rheological test [27,28]. The power law model is well-known for its extensively utilization in describing the flow properties of non-Newtonian liquids. The obtained rheological data were fitted to the power law model (Equations (1)–(3)) to investigate the variation in the rheological properties of set yoghurts added with different polysaccharides under steady shear and oscillatory sweep.
(1)$$\eta =K\gamma^{n}$$
(2)$$G\prime =K\prime f^{n\prime }$$
(3)$$G\prime \prime =K\prime \prime f^{n\prime \prime }$$

In Equations (1)–(3), *η*, G′, and G″ represent the apparent viscosity (Pa·s), elastic modulus (Pa), and viscous modulus (Pa), respectively. *K*, *K*′, and *K*″ represent the consistency index, while *γ* and *f* represent the shear rate (s^−1^) and frequency (Hz), respectively. The *n*, *n*′, and *n*″ all represent the flow behavior index (dimensionless). The apparent viscosity (η), elastic modulus (G′), viscous modulus (G″), and loss tangent (tanθ = G″/G′) were obtained by TA rheometer data analysis software.

### 2.6. Microstructure Observation by Cryo-SEM and CLSM

The yoghurt samples were prepared for cryo-SEM by mounting them onto copper holders and plunging into liquid nitrogen slush at −210 °C [29]. The frozen samples were transferred to the cryo-preparation chamber, and quick sectioning was performed with a cold scalpel blade at −140 °C. The specimen was then etched at −90 °C for 5 min and coated with 300 Å of sputtered gold. The specimen was transferred under vacuum onto the cold stage, maintained at −95 °C, and imaged using SEM (FEI, Quanta 600 F, New York, NY, USA) at 5 kV.

Based on the method of Laiho et al. [30], the Fast Green FCF (0.1 mg/mL, McLean) was set as fluorescent stain to label protein and added to mixed with the yoghurt samples thoroughly. The prepared sample was dripped on a slide and covered with a cover slip, which were observed by UltraVIEW VoX 3D living cell laser confocal microscope imaging system (Perkin, Emer, Germany) equipped with 40× objective lens at the excitation wavelengths of 633 nm.

### 2.7. Statistical Analysis

All the obtained data based on experimental design were expressed as means ± standard deviation and analyzed by one-way ANOVA statistically method at the confidence level of *p* < 0.05 through IBM SPSS Statistics 25.0 software (SPSS Inc., Armonk, NY, USA). Graphs were constructed using Origin 2019b (Origin Lab, Northhampton, MA, USA). The experiments were all performed in triplicates.

## 3. Results and Discussion

### 3.1. Physical Properties, WHC, and Texture of Different Set Yoghurts

Compared with the control yoghurt, no significant difference was found among TA values of set yoghurts added with different concentrations of SA or GG except KGM (Table 1). This indicated that the additions of anionic polysaccharides have no effects on the trend of TA changes in set yoghurts, likely owing to the lack of their significant or adverse effect on the activity of fermented starter bacteria in yoghurts [1]. The pH values of set yoghurts added with different concentrations of SA or GG or KGM was significantly different (*p* < 0.05). This demonstrated that the addition of polysaccharides could make a certain impact on the pH value of set yoghurts, whether anionic or neutral polysaccharides [31,32]. Moreover, no obvious differences were found among the TSS of set yoghurts added with different concentrations or kinds, and both TS and TSS increased with increasing concentrations of added polysaccharides. Consistent observations were found by Nutthaya et al. [33] and Xu et al. [22] in set yoghurts with addition of inulin and okra polysaccharide, respectively.

The WHC measured by centrifugation method reflected the dehydration and stability of set yoghurts [34]. Higher WHC illustrated that less whey separated from the yoghurt through centrifugation. The WHC of the set yoghurts all presented a slight increase after adding 0.005% of SA, GG, and KGM, respectively. Additionally, the WHC of set yoghurts in the same group showed significant increase (*p* < 0.05) with the increasing quantity of added polysaccharides. In addition, the WHC of set yoghurt increased from 78.37% to 81.94% when the added concentration of GG was up to 0.02%. A similar trend was found in the set yoghurts added with SA or KGM concentrations ranged from 0.005% to 0.02%, the WHC of which increased from 78.21% and 78.01% to 80.85% and 81.01%, respectively. The above results indicated that the addition of polysaccharide could effectively and significantly improve the stability of set yoghurts. As Gyawali et al. [35] reported, this may be correlated with the hydrophile of polysaccharide molecules enhancing the rigidity of the protein gel network. Furthermore, it also may be related to the interaction between polysaccharides and protein molecules, particularly anionic polysaccharides, which could form complexes with protein clusters charged positively so as to improve the structure of protein gels [21,22,36]. 

The firmness of set yoghurt added with GG or KGM increased with increasing polysaccharide concentration, which could be correlative with the high WHC of set yoghurts added with polysaccharides [37,38]. Significant difference was also found among set yoghurts added with different concentrations of KGM (*p* < 0.05). For the set yoghurt added with GG, the firmness could increase from 1.17 N to 1.32 N when the addition was up to 0.02%, which was higher than that of set yoghurt added with the same concentration of SA and KGM. Additionally, the firmness of set yoghurt added with SA was similar to that of the control. The similar results were reported by Xu et al. [22], which may be related to the negative effect of SA leading to the formation of weak protein gel and the ratio of ᴅ-mannitonic acid and ʟ-mannose acid in SA structure as well. The similar phenomenon was found in the observation of an anionic polysaccharide-containing sulphate group—carrageenan. Carrageenan could induce the early gelation of milk at a low concentration (≤0.05%) but entirely inhibited gelation at the 0.2% concentration [39].

The rupture distance is another significant parameter for textural characterization of set yoghurt. The better cohesion of the gel was indicated by longer rupture distance [40]. The correlation between the rupture distance of set yoghurts and the added concentration of GG was more obviously positive than SA and KGM (*p* < 0.05). The rupture distance of set yoghurts added with 0.02% (*w*/*v*) GG were up to 3.48 mm, distinctly higher than that of set yoghurts added with 0.02% SA and KGM, and the corresponding rupture distance is 2.96 mm and 3.05 mm, respectively.

### 3.2. LF-NMR Analysis

The T_2_ and its corresponding peak area were used to reflect the mobility of water molecules containing hydrogen protons [11]. They are correlated with the extent of protein proton exchange, water proton exchange, and the interaction between water components and other polymers in gelation process [16,41]. The relative signal intensities (T_21_, T_22_, and T_23_) represent bound water, semi-bound water, and free water, respectively. Generally, three peaks were observed in the T_2_ distribution spectrum of set yoghurt added with SA, GG, and KGM (Figure 1). The first peak (T_21_) between the shortest relaxation time of 3.0 ms−6.0 ms corresponded to the bound water in set yoghurts, and the second peak (T_22_) was located in 20.0–65.0 ms, corresponding to non-flowing water in set yoghurts, which both have little effect on water-holding capacity and gel strength [21]. Finally, the third peak (T_23_) was found within the longest relaxation time of 300 ms, which was ascribed to the amount of free water [13]. Besides, the addition of SA, GG, and KGM all presented no obvious effect on T_21_ and T_22_. The T_23_ of set yoghurts added with SA, GG, and KGM did not change significantly compared with that of the control, indicating that the addition of polysaccharides basically had no effect on the water molecular migration rate in fermented milk.

The free water area (A_23_) reflects the free water-holding capacity of protein gel structure [22]. The decreased tendency of A_23_ value and significant difference to the control (*p* < 0.05) were all found in the set yoghurts added with three different kinds of polysaccharides. The decrease of A_23_ in set yoghurts added with GG was the largest compared to the set yoghurts added with SA and KGM. The results indicated that GG has better ability to promote gel structure hydration and reduce water flow in matrix space. Moreover, with the increased addition of SA, GG, and KGM, the A_23_ value of set yoghurts decreased compared to the control, which was consistent with the firmness result of set yoghurts (Table 1). This illustrated that the firmness of set yoghurts could be influenced by free water content and in accordance with the previous finding [17].

### 3.3. Rheological Analysis

The set yoghurts added with SA, GG, and KGM presented different apparent viscosity (Figure 2). The apparent viscosities decreased obviously as the shear rate rose. All the fermented set yoghurts showed shear-thinning behavior and pseudoplasticity phenomenon, which corresponds to the results of Xu et al. [22] about the apparent viscosity of set yoghurts added with okra polysaccharides. Based on the report of Cui et al. [28], yoghurt shows the characteristics of pseudoplastic phenomenon and shear-thinning behavior mainly related to the breakage of bonds between protein aggregation, which is conducive to comparing the differences of rheological properties between different yoghurt samples. Compared with the control, when the concentration of added SA, GG, and KGM was 0.005%, the apparent viscosity of the set yoghurts presented little increase in different degrees, which was related to the bridging effect of polysaccharide under low concentration. Then, with the added concentration of SA, GG, and KGM further increasing, the apparent viscosity increased correspondingly, indicating that the internal structure of set yoghurts tended to be denser and more uniform.

The frequency sweep curves of all set yoghurts added with SA, GG, and KGM were shown in Figure 3. In the frequency range of 0.1–10 Hz, the elastic modulus (G′), and viscous modulus (G″) of all set yoghurts presented an upward tendency, and the logarithm of G′ was always higher than G″, indicating that all set yoghurts had good elasticity and gel structure. The larger the elastic modulus value, the stronger the interaction between particles and the more stable the network structure [28]. It is obvious that as the concentration of SA, GG, or KGM addition increased, an increase in the G′ and G″ of the set yoghurts was observed, which corresponded to the apparent viscosity as well (Figure 2). With the increase of GG addition from 0.005% to 0.02% (Figure 3B), the G′ logarithm of set yoghurts presented a significant increase with the addition of 0.02% GG, and the results indicated that polysaccharides could produce marked effects on the improvement of rheological properties in set yoghurts and could be closely related to the types of polysaccharide stabilizers [10].

The apparent viscosity (*η*) versus shear rate (*γ*) data at 4 °C were well-fitted to the simple power law model (Equation (1)) with high correlation coefficients (R^2^ = 0.999), as shown in Table 2. There was no significant difference of the *n* value among the set yoghurts added with different concentrations of SA, GG, and KGM, indicating that the pseudoplastic fluid properties of the set yoghurts did not change significantly with the addition of the three types of polysaccharide stabilizers. The values of consistency index (*K*) obtained from the power law model (Equation (1)) increased when the added concentration of SA, GG, and KGM increased from 0.005% to 0.02%. For the one polysaccharide, the corresponding *K* value of set yoghurts added with different concentrations of polysaccharide presented a significant difference (*p* < 0.05). When adding the same concentration of polysaccharides, the *K* value of set yoghurts added with GG was significantly higher than that added with SA and KGM (*p* < 0.05). Taken together, it was found that the set yoghurts added with concentrations of SA, GG, and KGM had higher shear-thinning behavior, and their steady shear properties were apparently influenced by the concentration of polysaccharide stabilizers addition. Further, the dynamic rheological data of log (G′, G″) versus log *f* were subjected to linear regression, the magnitudes of the slopes (*n*′) and (*n*″), consistency index (*K*′ and *K*″), and R^2^ in the Equations (2) and (3). The values of *K*′ and *K*″ increased with an increase in SA, GG, and KGM concentration, and *K*′ values were much larger than those of *K*″. Similar to the change of *K* value, for the one polysaccharide, the corresponding *K*′ and *K*″ values of set yoghurts added with different concentration of polysaccharide presented significant difference as well (*p* < 0.05). When the concentration of the three polysaccharides was 0.02%, the *K*′ and *K*″ values of the set yoghurts added with GG were significantly different from that added with SA and KGM (*p* < 0.05). Additionally, the slopes of set yoghurts added with SA, GG, and KGM with *n*″ were relatively lower than *n*′, which indicated that the elastic properties of set yoghurts may be increased by the addition of polysaccharide stabilizers [42].

As the ratio of G″ to G′, the loss tangent (tanδ) was as a function of frequency to better compare the effect of SA, GG, and KGM addition on the gel structure of yoghurt (Figure 4). The lower the tanδ value was, the less susceptibility to syneresis the yoghurt was and the weaker the dehydration and remodeling was [21,43]. The tanδ of all set yoghurts were all below 1, suggesting the good gel texture of all the set yoghurts. In addition, the tanδ of set yoghurts added with SA and GG showed significant decrease with their increasing addition and remained stable during frequency sweep, illustrating that the addition of polysaccharides enabled to effectively reduce yoghurt syneresis. Noticeably, no significant difference was found in the minimum tanδ value of set yoghurts added with SA, GG, and KGM, which was 0.21, 0.22, and 0.23, respectively. This indicated the yoghurt added with the three polysaccharides had a higher level of elasticity, and it was not easy for the whey to be dehydrated and precipitated [44]. However, the tanδ of set yoghurts added with KGM almost overlapped at the concentration of 0.005%, 0.010%, and 0.015% and then presented an obvious decrease when the set yoghurts were added with 0.020% KGM, corresponding to the results of frequency scanning as before (Figure 3). This demonstrated that yoghurts added with different types of polysaccharides presented different structure changes and could be closely related to types of polysaccharides. Additionally, Zhi et al. [45] and Xu et al. [22] also reported that tanδ with temperature sweep could reflect the viscoelastic modulus changes during cooling–melting cycles for hydrogels. 

### 3.4. Microstructure Observation and Comparison

The microstructures of set yoghurts added with SA, GG, and KGM were observed by cryo-SEM (Figure 5). Three types of set yoghurts showed a porous, web-like structure. The gel network of set yoghurts added with SA (Figure 5a) had a larger pore size than that added with GG and KGM (Figure 5b,c). Water was sublimated more rapidly in set yoghurt added with SA or KGM compared to that added with GG. Yoghurts added with GG were likely to contain more water owing to the high capillary force of small pores. Similarly, a number of studies reported that produced porous structures could be advantageous for obtaining lower syneresis in yoghurts [10,18,46]. Besides, owing to the more interconnected gel networks and denser pore structure, the firmness and WHC of the set yoghurts added with GG may be higher. Moreover, the filamentous gel chains are replaced by aggregates in the yoghurts added with GG, and aggregates are joined together to form a stronger network structure.

The difference of microstructure in set yoghurts added with SA, GG, and KGM were obtained by CLSM observation (Figure 6). The spacing gaps occupied by whey (black) and the casein clusters (red) were evenly distributed. The density of casein clusters increased, and the space between casein clusters became smaller in set yoghurts added with GG compared with the control (Figure 6a). When the concentration of GG added were highest 0.02%, a greater aggregation of casein clusters was observed (Figure 6b). This could be correlative with the inhibition of the cross gap effect of polysaccharides on the connection between casein clusters [47]. The caseins cluster structure of set yoghurts added with SA presented an obvious difference compared to that of set yoghurts added with GG (Figure 6c). After adding 0.005% SA, the protein network of set yoghurt was loose, and the gap was large, the casein clusters were dispersed, and the connection between groups was weak. Similar results were reported by Tudorica et al. [48], who stated that the coagulation effect of the addition of *β*-dextran is consistent with that of SA, which could expand the pores and form a more open network structure. Additionally, Lee et al. [49] and Zhao et al. [50] reported that the weakening of cross-linking between micelles were the reason for the poor WHC and firmness of the samples. This corresponded to the analysis on WHC and firmness in the previous physicochemical analysis (Table 1). With the increase of SA concentration, the structure of the casein network is more intensive, which helps to increase the elasticity, WHC, and firmness of the gel (Figure 2 and Table 1).

The interaction of casein micelles in set yoghurts added with neutral polysaccharide KGM was slightly weaker compared to that in set yoghurts added with acidic polysaccharide GG. When the concentration was 0.005%, there was no significant effect on the protein network structure (Figure 6d). Set yoghurts added with KGM showed smaller protein clusters and looser connections compared to the set yoghurts added with GG in microstructure. However, when the addition of KGM was further increased to 0.01%, 0.015% and 0.02%, relatively more and more larger protein clusters were formed. This may be owing to the strong hydration ability of KGM, which could promote the aggregation of casein micelles during the acidification process and reduce the flexibility of the gel structure [51,52,53]. A similar effect was observed in fermented milk with natural starch, as reported by Pang et al. [39]. In general, with the gradual increase of KGM concentration, the WHC, elasticity, and apparent viscosity of prepared set yoghurts were improved to a certain extent.

Based on the results observed above, the effect mechanism of different types of polysaccharides on the gel structure was illustrated by a schematic model, as shown in Figure 7. In the fermentation process, protein aggregation and protein–polysaccharide interaction include electrostatic attraction and other physical interactions, such as van der Waals force, hydrophobic force, volume exclusion, steric repulsion, and entropy effects, etc. [39]. The electrostatic attraction is the most important one; that is, the interaction between positively charged casein micelles and negatively charged anionic polysaccharides (SA and GG) could form complexes by the COO^−^ group of anionic polysaccharides and NH^3+^ group of casein. According to the concentration and structure of polysaccharides, complexation can reduce or enhance the gel strength. Among them, GG has high viscosity and contains a large number of negatively charged residues, which can significantly promote the strength and structure of set yoghurt. Moreover, the phthalein in a neutral polysaccharide KGM molecule can bind to the CH_3_ of casein to form a hydrogen bond [54], but its interaction is relatively weak. It may only play a filling role in the gel network and could increase the viscosity of the continuous phase.

## 4. Conclusions

This study investigated and compared the effects of three common commercial polysaccharide stabilizers (SA, GG, and KGM) on the textural, rheological, and microstructural characteristics of set yoghurts. The results showed that the addition of GG obviously enhanced the WHC, firmness, cohesion, and elasticity of set yoghurts. The microstructure of set yoghurt added with GG presented denser and more uniform protein clusters and smaller porous structure compared with the control and that of set yoghurts added with SA or KGM. The addition of neutral polysaccharides KGM increased the cross-linking of casein micelles slightly during fermentation process as well, whereas the effect of promoting the cross-linking of casein micelles in set yoghurts added with SA was relatively weakest. In a word, compared with other two polysaccharides, the excellent effect on improving the texture and structure of set yoghurts added with GG was demonstrated, which indicated that maybe microbial anionic polysaccharides have better potential to be utilized as the stabilizer and modifier for textural properties of dairy products. 

## Figures and Tables

**Figure 1 foods-11-01764-f001:**
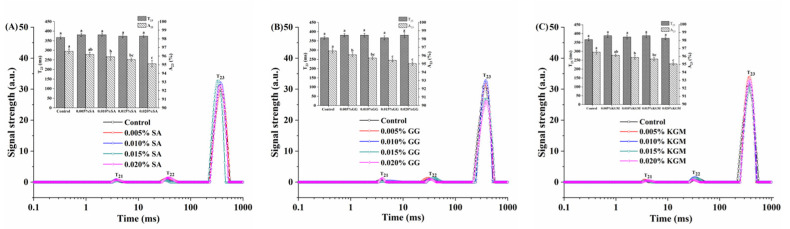
Distribution of T_2_ relaxation times on set yoghurts at different polysaccharides concentrations by LF-NMR. (**A**) SA, (**B**) GG, and (**C**) KGM. SA, sodium alginate; GG, gellan gum; KGM, konjac glucomannan. The different letters (e.g., a, b, …) mean significant difference of the longest relaxation time(T_23_) and free water area(A_23_) among set yoghurts with different added concentrations of the one polysaccharide (*p* < 0.05).

**Figure 2 foods-11-01764-f002:**
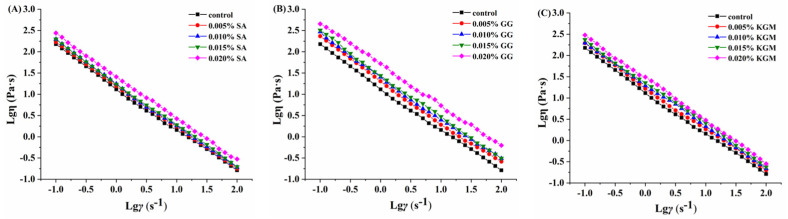
Logarithmic diagram of apparent viscosity changes of set yoghurts with different polysaccharide addition: (**A**) SA, (**B**) GG, and (**C**) KGM.

**Figure 3 foods-11-01764-f003:**
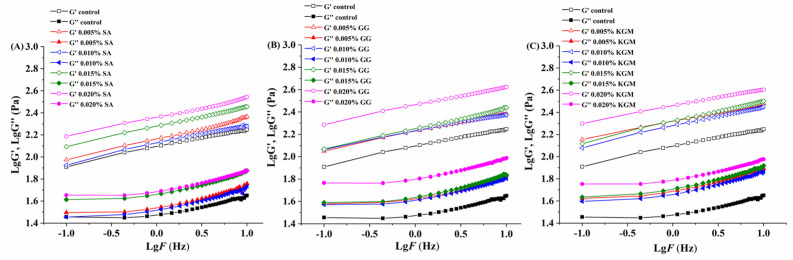
Logarithmic diagram of elastic modulus G′ and viscous modulus G″ as a function of frequency for set yoghurts added with different polysaccharides concentrations: (**A**) SA, (**B**) GG, and (**C**) KGM.

**Figure 4 foods-11-01764-f004:**
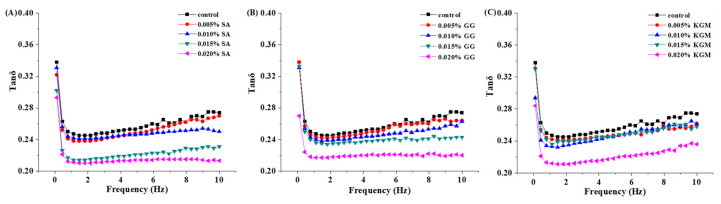
Tanδ as a function of frequency for set yoghurts added with different polysaccharides concentrations: (**A**) SA, (**B**) GG, and (**C**) KGM.

**Figure 5 foods-11-01764-f005:**
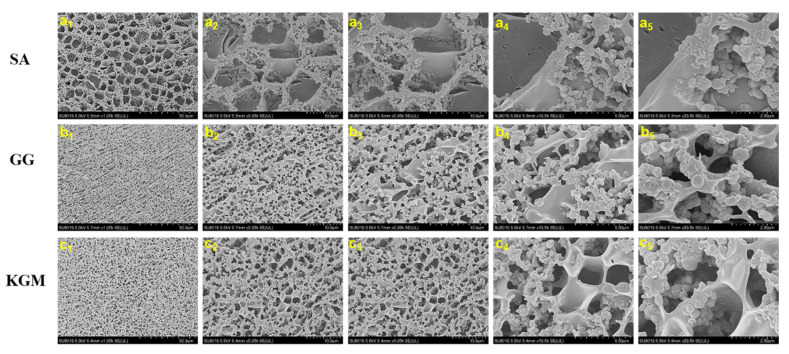
Cryo-SEM micrographs of set yoghurts added with SA (**a**), GG (**b**), and KGM (**c**) at different magnifications of 1.00 k×, 3.00 k×, 5.00 k×, 10.00 k×, and 20.00 k×.

**Figure 6 foods-11-01764-f006:**
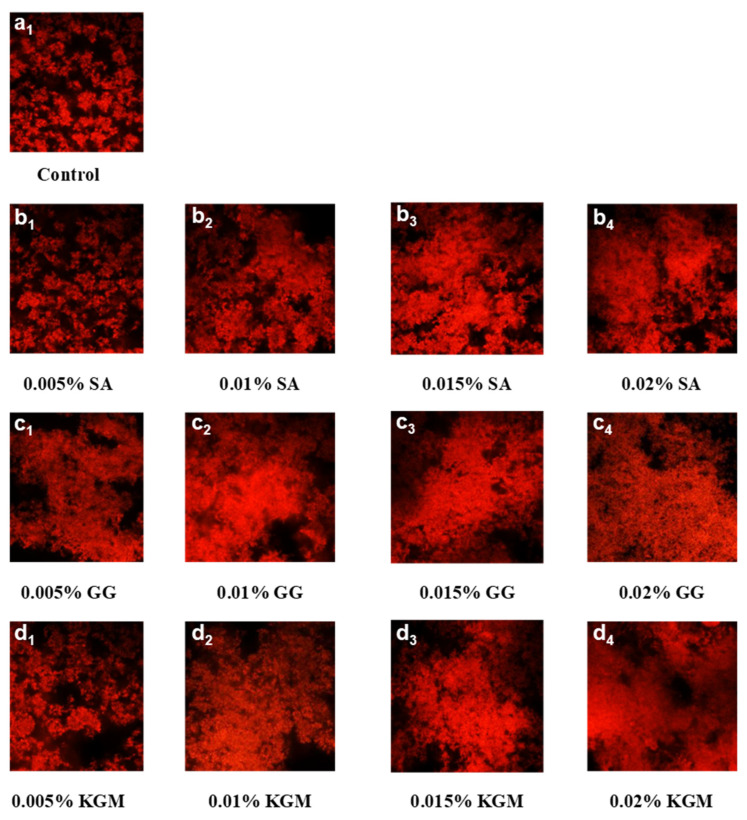
The CLSM micrographs of the protein network in the control set yoghurt with no addition (**a**), and the set yoghurts added with SA (**b**), GG (**c**), and KGM (**d**) at concentrations of 0.005%, 0.01%, 0.015%, and 0.02% (*w*/*v*).

**Figure 7 foods-11-01764-f007:**
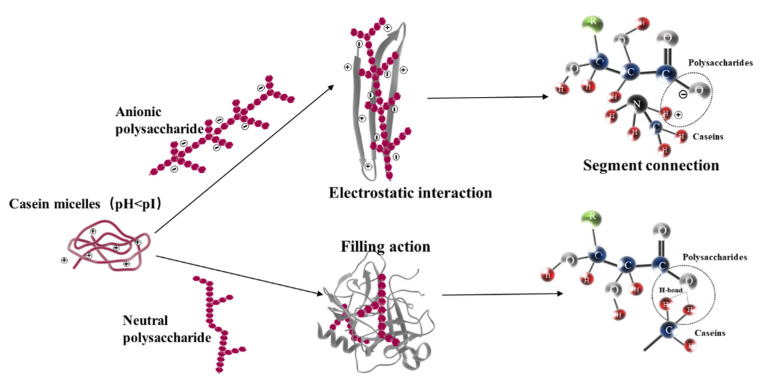
The description of the interaction between casein and polysaccharide in set yoghurts.

**Table 1 foods-11-01764-t001:** Effects of different polysaccharides stabilizers on physicochemical properties of set yoghurts.

Samples		Titratable Acidity (°T)	pH	TS (%)	TSS ^ns^ (°Brix)	WHC (%)	Firmness (N)	Rupture Distance (mm)
Control		96.79 ± 0.55	4.51 ± 0.03 ^c^	11.95 ± 0.09 ^b^	7.37 ±0.15	77.98 ± 0.89 ^c^	1.09 ± 0.08	2.23 ± 0.07 ^c^
SA	0.005%	96.84 ± 0.45 ^A^	4.64 ± 0.03 ^Aa^	11.99 ± 0.11 ^Bb^	7.31 ± 0.21	78.21 ± 0.57 ^bc^	1.11 ± 0.09	2.27 ± 0.07 ^Bc^
	0.010%	96.32 ± 0.33	4.60 ± 0.01 ^Aab^	12.07 ± 0.06 ^Bab^	7.35 ± 0.92	79.29 ± 0.32 ^b^	1.13 ± 0.10	2.39 ± 0.11 ^Bc^
	0.015%	96.73 ± 0.51 ^A^	4.55 ± 0.03 ^bc^	12.16 ± 0.05 ^Ba^	7.38 ± 0.23	80.14 ± 0.98 ^ab^	1.14 ± 0.11	2.76 ± 0.03 ^Cb^
	0.020%	96.71 ± 0.77	4.59 ± 0.02 ^b^	12.20 ± 0.03 ^Ba^	7.39 ± 0.10	80.85 ± 0.61 ^a^	1.12 ± 0.12	2.96 ± 0.13 ^Ba^
Control		96.79 ± 0.55	4.51 ± 0.03 ^b^	11.95 ± 0.09 ^c^	7.37 ± 0.15	77.98 ± 0.89 ^c^	1.09 ± 0.08	2.23 ± 0.07 ^e^
GG	0.005%	95.89 ± 0.62 ^AB^	4.56 ± 0.03 ^Ba^	12.21 ± 0.06 ^Ab^	7.38 ± 0.25	78.37 ± 0.13 ^c^	1.17 ± 0.11	2.75 ± 0.04 ^Ad^
	0.010%	96.12 ± 0.78	4.54 ± 0.01 ^Bab^	12.25 ± 0.10 ^Aab^	7.40 ± 0.01	79.87 ± 0.27 ^b^	1.21 ± 0.17	2.99 ± 0.02 ^Ac^
	0.015%	96.38 ± 0.41 ^AB^	4.57 ± 0.03 ^a^	12.29 ± 0.02 ^Aa^	7.43 ± 0.35	80.35 ± 0.45 ^b^	1.28 ± 0.35	3.21 ± 0.08 ^Ab^
	0.020%	96.06 ± 0.57	4.58 ± 0.04 ^a^	12.34 ± 0.04 ^Aa^	7.45 ± 0.12	81.94 ± 1.18 ^a^	1.32 ± 0.10	3.48 ± 0.15 ^Aa^
Control		96.79 ± 0.55 ^a^	4.51 ± 0.03 ^c^	11.95 ± 0.09 ^c^	7.37 ± 0.15	77.98 ± 0.89 ^c^	1.09 ± 0.08 ^b^	2.23 ± 0.07 ^c^
KGM	0.005%	95.21 ± 0.87 ^Bb^	4.56 ± 0.01 ^Bb^	12.01 ± 0.01 ^Bbc^	7.40 ± 0.45	78.01 ± 0.58 ^c^	1.14 ± 0.10 ^ab^	2.73 ± 0.01 ^Ab^
	0.010%	95.60 ± 0.59 ^b^	4.59 ± 0.01 ^Aab^	12.10 ± 0.04 ^Bb^	7.42 ± 0.95	79.54 ± 0.43 ^b^	1.16 ± 0.09 ^a^	2.82 ± 0.27 ^Aab^
	0.015%	95.71 ± 0.04 ^Bb^	4.61 ± 0.04 ^a^	12.23 ± 0.05 ^ABa^	7.45 ± 0.31	79.87 ± 0.40 ^b^	1.20 ± 0.09 ^a^	2.97 ± 0.06 ^Ba^
	0.020%	95.58 ± 0.28 ^b^	4.54 ± 0.02 ^bc^	12.28 ± 0.03 ^Aa^	7.47 ± 0.26	81.01 ± 0.61 ^a^	1.26 ± 0.08 ^a^	3.05 ± 0.12 ^Ba^

Note: ns, not significantly different (*p* > 0.05). The different letters (e.g., a, b, …) in the same column mean significant difference among set yoghurts with different addition concentrations of the one polysaccharide (*p* < 0.05). The different letters (e.g., A, B, …) in the same column mean significant difference among set yoghurts added with different kinds of polysaccharides under the one concentration. SA, sodium alginate; GG, gellan gum; KGM, konjac glucomannan. TS, total solids; TSS, total soluble solids; WHC, water-holding capacity.

**Table 2 foods-11-01764-t002:** Parameters of set yoghurts added with different polysaccharides stabilizers estimated by Power Law equation.

Sample		*K*	*n* ^ns^	R^2^	*K*′	*n*′ ^ns^	R^2^	*K*″	*n*″ ^ns^	R^2^
Control		14.09 ± 0.27 ^e^	0.98 ± 0.03	0.999	123.20 ± 6.98 ^d^	0.16 ± 0.08	0.993	31.54 ± 1.08 ^d^	0.12 ± 0.03	0.859
SA	0.005%	16.25 ± 0.13 ^Cd^	0.98 ± 0.05	0.999	144.64 ± 8.63 ^Bc^	0.19 ± 0.01	0.993	36.65 ± 1.16 ^Cc^	0.15 ± 0.01	0.888
	0.010%	17.85 ± 0.21 ^Cc^	0.99 ± 0.06	0.999	132.25 ± 9.13 ^Ccd^	0.17 ± 0.09	0.991	34.58 ± 1.21 ^Cc^	0.16 ± 0.04	0.913
	0.015%	18.39 ± 0.14 ^Cb^	0.98 ± 0.10	0.999	188.80 ± 7.86 ^Bb^	0.18 ± 0.06	0.999	48.51 ± 1.56 ^Bb^	0.16 ± 0.08	0.897
	0.020%	26.27 ± 0.32 ^Ba^	0.98 ± 0.13	0.999	227.51 ± 6.88 ^Ba^	0.17 ± 0.10	0.993	51.38 ± 1.79 ^B^^a^	0.14 ± 0.05	0.885
Control		14.09 ± 0.27 ^d^	0.98 ± 0.03	0.999	123.20 ± 6.98 ^c^	0.16 ± 0.08	0.993	31.54 ± 1.08 ^c^	0.12 ± 0.03	0.859
GG	0.005%	20.93 ± 0.28 ^Ac^	0.99 ± 0.05	0.998	167.88 ± 3.15 ^Bbc^	0.16 ± 0.08	0.997	44.39 ± 1.57 ^Bb^	0.15 ± 0.02	0.894
	0.010%	25.28 ± 0.24 ^Abc^	0.98 ± 0.14	0.999	169.51 ± 4.98 ^Bbc^	0.15 ± 0.04	0.997	43.27 ± 1.34 ^Bb^	0.15 ± 0.08	0.890
	0.015%	29.05 ± 0.16 ^Ab^	0.99 ± 0.09	0.999	177.30 ± 3.09 ^Bb^	0.18 ± 0.04	0.997	45.88 ± 1.42 ^Bb^	0.16 ± 0.05	0.900
	0.020%	50.59 ± 0.41 ^Aa^	0.96 ± 0.04	0.999	288.74 ± 6.08 ^Aa^	0.16 ± 0.05	0.998	66.76 ± 2.99 ^Aa^	0.14 ± 0.03	0.892
Control		14.09 ± 0.27 ^e^	0.98 ± 0.03	0.999	123.20 ± 6.98 ^d^	0.16 ± 0.08	0.993	31.54 ± 1.08 ^d^	0.12 ± 0.03	0.859
KGM	0.005%	18.02 ± 0.11 ^Bd^	0.98 ± 0.05	0.999	208.31 ± 6.46 ^Ab^	0.16 ± 0.09	0.999	51.48 ± 2.03 ^Ab^	0.16 ± 0.02	0.923
	0.010%	20.38 ± 0.19 ^Bc^	0.98 ± 0.02	0.999	187.97 ± 5.65 ^Ac^	0.18 ± 0.03	0.997	48.24 ± 1.98 ^Ac^	0.16 ± 0.05	0.923
	0.015%	23.07 ± 0.20 ^Bb^	0.97 ± 0.08	0.999	207.01 ± 7.82 ^Ab^	0.18 ± 0.02	0.998	53.48 ± 2.11 ^Ab^	0.16 ± 0.07	0.933
	0.020%	29.46 ± 0.23 ^Ba^	0.99 ± 0.05	0.999	287.67 ± 8.33 ^Aa^	0.15 ± 0.05	0.998	64.95 ± 2.54 ^Aa^	0.14 ± 0.06	0.891

Note: ns, not significantly different (*p* > 0.05). The different letters (e.g., a, b, …) in the same column mean significant difference among set yoghurts with different addition concentrations of the one polysaccharide (*p* < 0.05). The different letters (e.g., A, B, …) in the same column mean significant difference among set yoghurts added with different kinds of polysaccharides under the same concentration.

## Data Availability

Data are available upon request.

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
