# Peer review of "Effects of Commercial Polysaccharides Stabilizers with Different Charges on Textural, Rheological, and Microstructural Characteristics of Set Yoghurts"

_foods, 2022, doi:10.3390/foods11121764_

Round 1

Reviewer 1 Report

ENOUGH WELL WRITTEN ABSTRACT and correctly selected keywords

An interestingly written introduction, exploring the subject and validity of the research, sufficiently introduces the reader to the subject of research

2.1. I think that for the sake of clarity on the subject of the main raw material used in research, it would be reasonable to write from which mammal the milk comes from. In fact, I took it for granted that cow's milk was used, but when she saw Mongolia ... she had doubts.

table 1 sample "checks" requires correct formatting text no space before and after - / +

Figure 2 and 4 requires the unification of the maximum values for the graph b is 12 and in the remaining (a and c) the max is 10. Such a presentation misleads the potential reader and the existing differences are blurred.

I love the graphic interpretation of phenomena, interaction processes in scientific works. Figure 7 is wonderful. I suggest placing it in the so-called graphic abstract ... because it will significantly increase the readership of the work

Reviewer 2 Report

I read carefully the article entitled "Effects of commercial polysaccharides stabilizers with different charges on textural, rheological and microstructural characteristics of yoghurts". However, the following points are provided for the authors:

  • Introduction:

It is very important to write sentences in the introduction that illustrate the importance and innovation aspects of research. The introduction should be completely rewritten and none of the sentences used in the present format should be in the edited version. Because instead of introducing yogurt and milk proteins and the fermentation process and the inhibiting role of hydrocolloids in syneresis phenomena, you have to explain your new perspective on this issue. In the introduction, you should look at how to evaluate yogurt syneresis using NMR and SEM etc.

  • Section 2.1:

It is essential that the molecular weight, M/G ratio and viscosity of all three hydrocolloids used be fully introduced.

  • Section 2.2:

Line 94: Given the different viscosities of these three hydrocolloids, what was the basis for selecting the concentrations used in yogurt?

Line 98: Why was the time required for fermentation until the formation of yogurt gel the same for all samples? Isn't it that yogurt is formed at a pH of 4.6? Did all samples reach this pH after 6 hours of incubation? Why did not you measure the pH? Isn't the main reason for the syneresis and breaking of yogurt gel, the drop in pH to less than 4.6 or not reaching this pH?

  • Section 2.5:

Are strains of 10 to 100 s-1 sufficient to study the flow behavior of yogurt? What about 0.1 to 10 Hz frequency?

  • Section 3.1:

Mean comparison was not performed correctly. You should compare the average once between samples with the same concentration of different hydrocolloids and once between different concentration of each hydrocolloid alone. In this case, you will see that the differences are significant and the analysis and trends will change completely.

The results of this section do not provide any new analysis. Trends are completely predictable and there is no innovation or new perspective. Even for these predictable and repetitive results, no strong and good analysis is provided.

  • Section 3.3

Line 233 – 243: What is the difference between the samples containing the three hydrocolloids and what is the observed difference?

Fig 2 and 3: It is necessary to draw a graph logarithmically to understand the frequency dependence and logarithmic distance between modules. It is also necessary to report the fit of the data with the power model and its parameters.

  • Section 3.5:

Title 3.5 should be deleted and these sentences should be merged in the results of other sections. In addition, in the research, you must prove the hypotheses you mentioned in the introduction using the tests performed. Hypotheses or conclusions beyond the tests performed should not be included in the article.

Round 2

Reviewer 2 Report

Please refer to the articles that have done the dynamic rheology test and consider how the diagrams are displayed and adjust them correctly.

How are negative values obtained for n values? You have definitely done some of the work wrong.

Unfortunately, the values presented in the pH and TS columns of Table 1 are incorrectly written.

Lines 130 to 133: The added items have a basic scientific objection.

The method of rheological measurements and flow behavior is incomplete.

The strains 10 to 100 s-1 are not sufficient to study the flow behavior of yogurt? It is true for 0.1 to 10 Hz frequency, too. Please check.

The results of the rheological test results are very poorly written. Be sure to rewrite it completely and state the results and reason for the observations accurately.

Author Response

Thank you very much for your comment. Please see the attachment.
